# Latent World Models For Intrinsically Motivated Exploration

**Aleksandr Ermolov, Nicu Sebe**
Department of Information Engineering and Computer Science (DISI)
University of Trento, Italy
{aleksandr.ermolov,niculae.sebe}@unitn.it

## Abstract

In this work we consider partially observable environments with sparse rewards. We present a self-supervised representation learning method for image-based observations, which arranges embeddings respecting temporal distance of observations. This representation is empirically robust to stochasticity and suitable for novelty detection from the error of a predictive forward model. We consider episodic and life-long uncertainties to guide the exploration. We propose to estimate the missing information about the environment with the world model, which operates in the learned latent space. As a motivation of the method, we analyse the exploration problem in a tabular Partially Observable Labyrinth. We demonstrate the method on image-based hard exploration environments from the Atari benchmark and report significant improvement with respect to prior work. The source code of the method and all the experiments is available at `https://github.com/htdt/lwm`.

## 1 Introduction

A sparse reward signal is one of the challenging problems in reinforcement learning (RL). In this case, random exploration is inefficient, the number of randomly performed steps grows exponentially with the number of distinct visited states, the agent is unlikely to stumble on the rewarding state. The intrinsic motivation, or curiosity, approach aims at this problem [4]. This method provides an intrinsic reward signal to encourage an agent to seek novel or rare states. There are several approaches to detect such states and calculate the reward: one class of methods estimates the novelty proportionally to an error of future state prediction (predictive forward models) [31, 25, 6], another class approximately counts visited states and estimates the novelty in an inverse ratio [32, 4, 10, 24, 26, 33]. Most of the methods estimate novelty in a life-long time frame, states are considered novel only if the agent did not observe them during the whole training. Some methods use an episodic time frame comparing only the states inside a specific episode [26, 2].

Often RL environments are only partially observable and the agent must maintain a belief state, i.e., a sufficient statistic of the past, required for action selection. For instance, in the Atari Pong game, the state should include at least two consecutive frames of the environment, making it possible to determine the direction of the ball movement. This is a common situation, when the environment is partially observable and the reward is sparse at the same time. For instance, consider a labyrinth composed of rooms similar to each other. The agent observes only the room in which it is currently located, the reward is given only when the agent reaches the goal. Our experiments (Tab. 1) show that even the 16 room labyrinth is a challenging task for a common RL algorithm. Novelty estimation, based on current observation, is also inefficient: each room looks similar even during one episode.

Visiting states with high uncertainty can be beneficial for exploration. We distinguish three types of uncertainty by sources: from partial observability, from novelty (epistemic) and from stochasticity (aleatoric). The latter type is irrelevant to the exploration strategy, whereas the first and the second

indicate states required to explore. We use a world model to estimate the missing information, i.e., a Recurrent Neural Network (RNN) trained to predict the next state given a current state and action, where a high prediction error indicates interesting states that may improve knowledge of the environment. The model maintains a belief state [11] allowing to detect missing information related to partial observability, e.g. prediction error changes before and after visiting a room during one episode. Being a predictive forward model, it allows to detect novel states in general, since the prediction error changes before and after the model is trained on the observation.

The desired properties of the world model for exploration are the insensitivity to stochasticity and the ability to extrapolate the state dynamics, such that the prediction error can be a measurement for novelty. We hypothesize that the temporal distance between observations indicates how different (and possibly novel) they are. We propose a self-supervised representation learning method based on minimization of Euclidean distance between representations of temporally close observations and the feature whitening to prevent degenerate solutions. The method produces a low-dimensional latent representation which is empirically robust to stochastic elements and is arranged respecting the temporal distance of observations. The proposed world model operates with latent representations, and the reconstruction error is calculated for low-dimensional latent states during the training, without relying on generative decoding of observation images.

To learn the exploration policy, we sum the environment (extrinsic) reward with the intrinsic reward proportional to the prediction error of the world model. Even if the initial environment was fully observable, the updated reward assignment transforms it into partially observable, i.e., the reward of a specific state may change during the episode. To address partial observability in our experiments we use a RNN architecture combined with a DQN (recurrent DQN) [13, 18]. Being off-policy, the algorithm maintains a replay buffer and reuses the past experience for training. We propose to recalculate the intrinsic reward for each sampling from the buffer, thus the estimation is always up-to-date, improving the sample efficiency of the algorithm. The experience from the buffer is also used to train the self-supervised representation and the world model.

Our contribution is as follows. We propose a method to estimate missing information and novelty in the environment with the world model, and we demonstrate it in the tabular Partially Observable Labyrinth. We introduce the self-supervised representation learning method that scales up the world model to environments with image observations. We compare our method with other common representation learning methods, demonstrating arrangement properties. We report the scores on several challenging Atari environments [3] with sparse rewards, improving results with respect to prior exploration methods.

## 2 Background and Related works

### 2.1 Partially Observable Markov Decision Processes

We consider a setting in which an agent interacts with an environment to maximize a cumulative reward signal. The interaction process is defined as a Partially Observable Markov Decision Process (POMDP) $< S, A, T, R, \Omega, O, \gamma >$ [23, 17], where $S$ is a finite state space, $A$ is a finite action space, $R(s, a)$ is the reward function, $T(\cdot|s, a)$ is the transition function, $\Omega$ is a set of possible observations, $O$ is a function which maps states to probability distributions over observations and $\gamma \in [0, 1)$ is a *discount factor* used to compute the cumulative discounted reward. Within this framework, at any given moment, the agent does not have access to the real state $s \in S$ of the environment, instead, it receives the observation $o \in \Omega$ with some incomplete information. After the agent takes an action $a \in A$, the environment transitions to state $s'$ with probability $T(s'|s, a)$, providing the agent with new observation $o' \sim O(s')$ and reward $r \sim R(s, a)$.

To overcome the short-term partial observability, Mnih et al. [22] represented the state as a current observation combined with 3 previous observations. Hausknecht and Stone [13] included a recurrent layer LSTM [14] to the model, thus the agent can summarize previous observations during the episode in the hidden state of the recurrent layer additionally to the current observation. Kapturowski et al. [18] have improved the performance of this algorithm proposing to sample long unrolls from the reply buffer and to collect the experience with distributed actors, we employ this algorithm for our experiments.

## 2.2 Exploration with intrinsic motivation

Several prior works employ predictive forward models on latent representations to estimate novelty. Stadie et al. [31] utilized an autoencoder representation. Pathak et al. [25] proposed to predict features obtained from an inverse dynamics model. Burda et al. [5] compared several latent representations: plain pixels, random features, variational autoencoder (VAE) [20] and inverse dynamics features. Our predictive forward model maintains a belief state to detect the missing information related to partial observability in addition to life-long novelty. Also, our method optimises the latent representation to follow the temporal alignment of observations.

Houthooft et al. [16] proposed a method based on probabilistic modelling of the environment and training the agent to maximize the information gain of actions. The method represents the agent's belief of the environment with parameters of a Bayesian neural network and directly measures the uncertainty. However, the method was not demonstrated on complex environments with image-based observations.

Kim et al. [19] have introduced latent representations of observations and actions, preserving linear topology in the representation space, consequently organizing functionally similar states close to each other. The method simultaneously arranges observations in a very low dimensional space and detects novelty from the arrangement error. This error is not always related to novelty; the method introduces an additional dynamics model to compensate it. Similarly, our method optimizes the latent representation to capture relationships of observations, although we separate representation learning from novelty estimation.

Savinov et al. [26] have considered an episodic time frame for novelty detection and proposed to use the temporal distance between observations to estimate the similarity. During an episode, the agent collects embeddings of observations in a buffer and compares new observations with previous ones, rewarding distant observations. The method relies on visual similarity, e.g. different rooms with the same appearance will be counted as one room.

Badia et al. [2] have reported high scores across both hard exploration and remaining games in the Atari-57 suite. The presented method for exploration combines a life-long novelty module (forward model for random features [6]) and an episodic novelty module. The latter encodes observations with an inverse dynamics model [25] and collects resulting embeddings in an episodic buffer, similar to [26]. To estimate novelty, the method calculates the Euclidean distance between a new embedding and its neighbors from the buffer. The main focus of the paper is a joint large scale (35 billion frames of each environment) training of multiple policies with different exploration and exploitation tradeoffs.

Sekar et al. [28] proposed to estimate novelty with a disagreement of an ensemble of predictive forward models inside a world model. The exploration is performed in a simulation, relying on a powerful world model that can generalize to novel unseen states. Our method estimates the novelty retrospectively, i.e., the world model is directly used to detect missing information. We employ a simple world model that operates with low dimensional latent states and it does not rely on generative decoding of the observation.

## 2.3 Self-supervised state representation learning

Unsupervised and self-supervised state representation learning methods aim to extract useful representations from state images without annotation and reward signal. One class of methods rely on generative decoding of the images. These methods minimise the reconstruction error in the pixel space, encouraging the representation to capture pixel-level details. Ha and Schmidhuber [12] combine VAEs with RNNs in their World Models. VAE provides a compact latent vector representation of each observation and the recurrent network represents the model of the environment inside this latent space. This network predicts the next latent vector from the current latent vector and an action, maintaining a hidden state. Similarly to this work, in our method, the RNN models the latent space. However, our latent representation is designed specifically for the exploration task and the model is used directly to estimate the prediction error.

Predictive State Representations [30] provide a theoretically grounded alternative to the World Models and the POMDP formulation in general. The method represents the state of the environment from a history of actions and observations as a set of predictions about the outcome of future tests.

The other class of methods is based on mutual information maximization between the representations of similar elements. Pair of images, sharing the same semantic content, is contrasted with other samples from the batch. van den Oord et al. [34] have introduced the Contrastive Predictive Coding (CPC) method, which optimizes the representation to encode information required to predict next frames of the sequence. Anand et al. [1] have proposed to maximize the mutual information between representations of two consecutive frames of the environment. The method relies on the global and patch-level representation of the image, capturing spatiotemporal factors. The method has been applied to several games from Atari 2600 suite, evaluating obtained representations in terms of disentanglement.

Our representation learning approach is conceptually similar to Slow Feature Analysis (SFA) [36], this method learns temporally invariant or slowly varying low-dimensional features from a stream of input data. The gradient-based extension of SFA [27] was demonstrated on image-based observations, this method can be an alternative to the proposed training procedure 3.1.

Ermolov et al. [9] have introduced a different approach for representation learning of similar elements (i.e., elements sharing the same semantics). The proposed W-MSE loss *scatters* representations of the batch in a spherical distribution[1] and then *pulls* the predefined similar elements closer to each other, minimizing the Euclidean distance between representations. Since the prediction error is calculated with this distance, such alignment in the latent space is preferred; we adopt this loss for our method.

# 3 Method

## 3.1 Representation learning for exploration

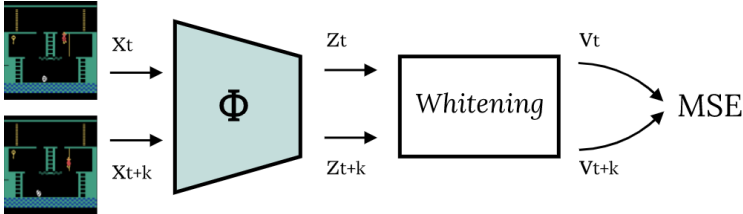

Figure 1: Representation learning procedure. Two temporally close frames are encoded with the network $\Phi$ and after whitening transform, the Euclidean distance between them is minimised with the MSE loss.

Latent representation plays the central role for the novelty estimation with the predictive forward models. The appropriate representation must filter out the stochasticity of observations (e.g. moving leaves of a tree in the wind) and should focus on the main scene elements (e.g. the controllable character in the game). Our method is based on the assumption that if the distance between compact latent representations of nearby frames is minimised, the stochastic elements will compensate each other, and at the same time, global trends will emerge. We use a convolutional neural network (CNN) with the architecture from [22] as an encoder and the output of the last layer is flattened and projected with a linear layer (i.e., the projection layer) to 32 dimensions. The encoder with the projection layer is denoted as $\Phi$. We denote frame index as $t$ and temporal shift as $k \in [1, L]$, they are sampled uniformly, frames $x_t$ and $x_{t+k}$ are queried from the reply buffer of the DQN agent. Maximum temporal shift $L$ defines which frames are considered similar; we use $L = 2$ in all the experiments. Following the idea of [21], we randomly shift both frames in a vertical and horizontal direction (0 - 4 pixels); this optional augmentation empirically regularises the representation stabilizing the training. The training procedure is depicted in Fig. 1 and is described below.

## 3.2 Whitening MSE Loss

After sampling two frames $x_t$ and $x_{t+k}$ as described in 3.1, which we call *positives* and denote $pos(x_t, x_{t+k})$, we obtain two corresponding vectors $\mathbf{v}_t = f(x_t, \theta)$ and $\mathbf{v}_{t+k} = f(x_{t+k}, \theta)$, where $f$ is a representation parameterized with $\theta$. Our goal is to minimize the distance between these vectors avoiding a degenerate solution (e.g. $f(\cdot, \theta) = constant$). We formulate this problem as:

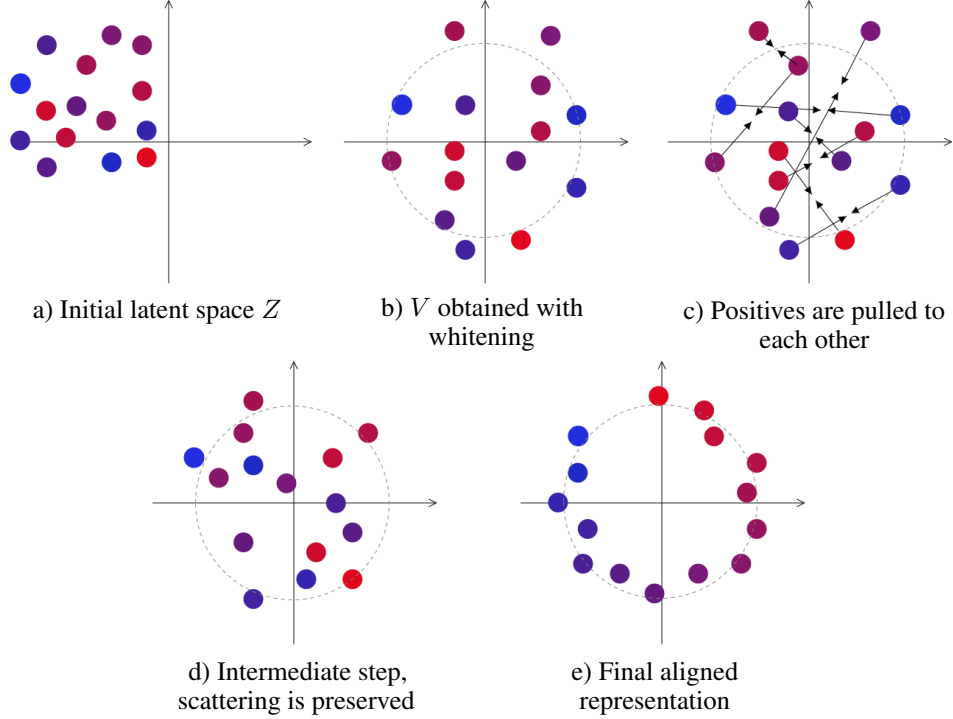

a) Initial latent space $Z$

b) $V$ obtained with whitening

c) Positives are pulled to each other

d) Intermediate step, scattering is preserved

e) Final aligned representation

Figure 2: Scheme of the optimisation process of the representation with the W-MSE loss. Circles denote latent vectors corresponding to observations, a similar color denotes temporally close observations. a) Initially, the representation is not structured. b) Elements are centred and scattered after whitening. c) MSE pulls positives to each other, while d) centring and scattering are preserved following Eq. (2). e) The final representation corresponds to the minimum of the target loss (3) placing consecutive elements sequentially.

$$min_\theta \, \mathbb{E} \, ||\mathbf{v}_t - \mathbf{v}_{t+k}||_2^2 \qquad (1)$$

$$s.t. \, cov(\mathbf{v}_t, \mathbf{v}_t) = cov(\mathbf{v}_{t+k}, \mathbf{v}_{t+k}) = I, \qquad (2)$$

where $I$ is an identity matrix. Eq. (2) constrains the distribution of $\mathbf{v}$ to be a spherical. We adopt the W-MSE loss [9] to optimise Eq. (1) and Eq. (2) is satisfied with the whitening transform [29]. We sample $N$ pairs of positives, obtaining a batch of $2N$ samples $B = \{x_1, ..., x_{2N}\}$. The corresponding vectors $Z = \{\mathbf{z}_1, ..., \mathbf{z}_{2N}\}$ are obtained from the encoder $\mathbf{z} = \Phi(x)$. We minimize the Mean Squared Error (MSE) over $N$ pairs after reparameterization of the $\mathbf{z}$ variables with whitened $\mathbf{v}$ variables:

$$L_{W-MSE}(V) = \frac{1}{N} \sum_{(\mathbf{v}_i, \mathbf{v}_j) \in V, pos(i,j)} ||\mathbf{v}_i - \mathbf{v}_j||_2^2, \qquad (3)$$

where $\mathbf{v} = Whitening(\mathbf{z})$, $Whitening(\mathbf{z}) = W_Z(\mathbf{z} - \boldsymbol{\mu}_Z)$, $\boldsymbol{\mu}_Z$ is the mean of the elements in $Z$, $\boldsymbol{\mu}_Z = \frac{1}{2N} \sum_n \mathbf{z}_n$, the matrix $W_Z$ is such that: $W_Z^\top W_Z = \Sigma_Z^{-1}$, being $\Sigma_Z$ the covariance matrix of $Z$, $\Sigma_Z = \frac{1}{2N-1} \sum_n (\mathbf{z}_n - \boldsymbol{\mu}_Z)(\mathbf{z}_n - \boldsymbol{\mu}_Z)^T$.

Each vector $\mathbf{z} \in Z$ is transformed with the whitening transform and the resulting set of vectors $V = \{\mathbf{v}_1, ..., \mathbf{v}_{2N}\}$ lies in a zero-centered distribution with a covariance matrix equal to the identity matrix (Fig. 2). For the whitening transform we adopt the efficient and stable Cholesky decomposition [8] proposed in [29]. For more details on the W-MSE loss we refer to Ermolov et al. [9], and for whitening transform to Siarohin et al. [29].

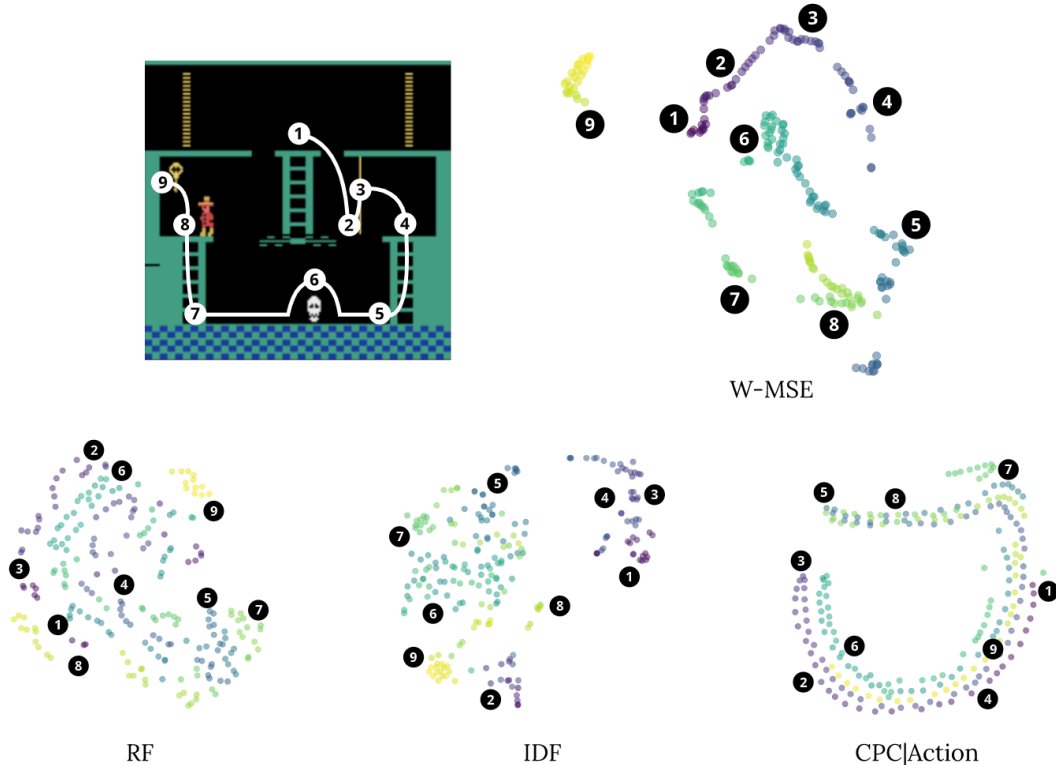

Figure 3: Trajectory visualisation in the latent space. Top-left image is a screenshot of the *Montezuma's Revenge* environment, white curve depicts the agent trajectory and numbers 1-9 match the corresponding points on visualisations. Each plot shows 32 dimensional embeddings of observations projected with the T-SNE method. The trajectory is created manually for demonstration, while representations are trained on random transitions.

## 3.3 Visual analysis of representations

To demonstrate the properties of the representation and to compare it with alternative common methods, we train methods on 400k observations of a random agent in *Montezuma's Revenge* environment and plot the obtained embeddings of one trajectory with the T-SNE [35] method in Fig. 3. We use $\Phi$ as the network architecture for all experiments.

Random Features (RF) [6] are embeddings produced with a network with randomly initialized and fixed weights. The intuition is that the architecture of the network itself produces the transformation, which is sufficient for suitable features. From the corresponding plot, we can see that the trajectory is structured, but the pattern is difficult to capture. Inverse Dynamics Features (IDF) [25] are obtained from a CNN trained on the prediction of action between two consecutive observations. Each observation is encoded with one network, two low dimensional outputs are concatenated and fed into a fully-connected network, the output dimension is equal to the number of actions. The intuition is that the network focuses on the elements that the agent can control, ignoring the rest. In our experiments, the action prediction accuracy after convergence was around 15% (random guessing is 6%) indicating that it does not always provide a strong training signal. The plot shows a better alignment compared to RF. However, several distant points are mixed as the network focuses on specific movements for action recognition.

CPC|Action [11] is an extension of the CPC [34] algorithm. A sequence of observations is processed with the CNN and the obtained embeddings are concatenated with the one-hot representation of actions. The sequence is aggregated with a RNN, and the resulting hidden state is used to predict the embedding of the next state of the sequence. The method is based on the InfoNCE loss [34], which requires the network to classify the positive pair (prediction and the target embedding) out of other

embeddings. From the plot, we can see that the network focused on repetitive animations, which are indeed valuable to contrast the frame from others, but are irrelevant to the exploration.

The W-MSE plot shows that states are arranged in curves, each curve corresponds to the semantic part of the trajectory. Similar states are pulled close to each other, and important events produce discontinuities in the trajectory. Such an arrangement simplifies the novelty estimation task and a very simple predictive forward model can capture these patterns.

### 3.4 Latent World Model

We employ [18] as an implementation of the recurrent DQN with some modifications: we use 1 frame as a state instead of 4; we do not decouple actors and learner, thus there is no lag in actors parameters update; we employ GRU [7] as RNN; full technical details are listed in the Supplementary.

The novelty and missing information are estimated as a prediction error of the world model. Our predictive forward model, which we call Latent World Model (LWM), operates with the representation obtained from the $\Phi$ encoder. The purpose of LWM is to approximate environment dynamics in the latent space. The encoding of the observation $z_t = \Phi(x_t)$ is concatenated with the one-hot action representation $a_t$, projected with a fully-connected layer and input to GRU, followed by two fully-connected layers. The network is optimised to predict the next encoding $z_{t+1}$ with the MSE loss.

The LWM and the recurrent DQN are trained jointly. During the warm-up phase, the experience is collected with a random policy and used to pretrain $\Phi$ and LWM (these steps are also included to the reply buffer and taken into account in the training budget). Next, for each training iteration, the unroll is sampled from the reply buffer and primarily processed with the LWM. The model produces the prediction error for each step of the unroll:

$$r_{i+1}^{intrinsic} = ||LWM(\Phi(x_i), a_i, b_i) - \Phi(x_{i+1})||_2^2, \tag{4}$$

where $i$ is a step of the unroll, $x_i$ is the observation, $a_i$ is the action taken in $x_i$, $b_i$ is the belief state of the LWN. We set $b_0 = 0$ for each unroll and the model performs 40 burn-in steps to obtain the actual starting belief state, following [18]. The error is backpropagated to update the weights of LWM and at the same time, it is used for novelty estimation. It is normalized with a running average of its mean and standard deviation; we use 0.999 momentum to update the running average. We clip the normalized value to range $[-10, 10]$ to remove outliers, and after this we multiply the resulting value with the coefficient $\beta$ to compensate for different magnitudes of intrinsic and extrinsic rewards. Finally, the resulting value for each step is summed with the extrinsic reward from the reply buffer and input to the recurrent DQN. Actors calculate LWM prediction error for the previous step and input it as a part of observation to the DQN. We adopt $\epsilon$-greedy action selection of [15], combining intrinsically motivated and random exploration policy with different $\epsilon$ for each actor. Note that the encoder $\Phi$ is updated with W-MSE loss during the whole training; empirically, the representation adapts to the changing distribution of observations. The algorithm and the configuration are available in the Supplementary.

## 4 Experiments

### 4.1 Partially Observable Labyrinth

We present a tabular environment with sparse rewards and partial observability. Each observation consists of 4 binary values, indicating the available doors on four walls in the square room. There are four possible actions: move up, move down, move left and move right. If there is a door in the chosen direction, the agent moves to a new room and receives a new observation. Otherwise, the agent stays in the same room and receives the same observation. At each step the agent receives the reward -1 and the episode finishes when all the rooms of the labyrinth are visited. Thus to get the highest cumulative reward, the agent must finish the episode as soon as possible, avoiding visiting the same rooms many times. At each new episode, the labyrinth is generated randomly with the recursive backtracker algorithm. The initial position of the agent is also sampled randomly. Thus the agent cannot memorize the layout and the solution requires to infer a simple navigation strategy. The agent does not have access to the full map of the labyrinth, current observation represents only the room where it is located (Fig. 4).

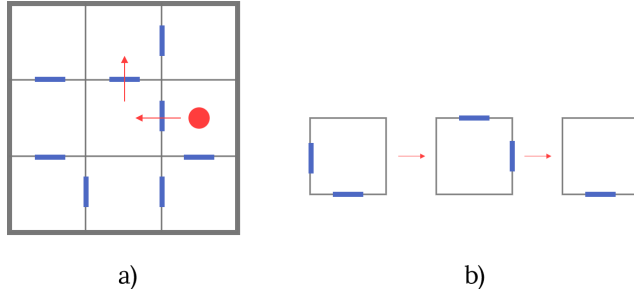

a)                                                    b)

Figure 4: Scheme of the Partially Observable Labyrinth. The red circle shows the starting position. In this example, the agent takes actions "left" and "up". a) demonstrates the whole labyrinth, this information is not available to the agent. Instead, it observes only a current room; sequence of the observations is depicted in b). Each observation consists of 4 binary numbers.

Table 1: Final cumulative rewards for Partially Observable Labyrinth. Episode length is limited to 1000. The recurrent DQN algorithm failed to solve the labyrinth starting from $4 \times 4$. The last column shows the results of the recurrent DQN with the world model for exploration, this method learned the optimal strategy for all sizes.

| Size | Random walk | Recurrent DQN | R. DQN + WM |
|------|-------------|---------------|-------------|
| $3 \times 3$ | -156 | -12.4 | **-10.8** |
| $4 \times 4$ | -518 | -1000 | **-23.5** |
| $5 \times 5$ | -848 | -1000 | **-72.1** |

The size of the labyrinth allows to regulate the complexity of the problem; this parameter is fixed during the training. We limit the length of the episode to 1000 steps. For evaluation, we average the cumulative reward over 128 different layouts of the labyrinth. We have performed experiments with 3 sizes of the labyrinth: $3 \times 3$, $4 \times 4$ and $5 \times 5$. Table 1 shows the performance of three agents: random, recurrent DQN and recurrent DQN with the world model for exploration. We estimate the intrinsic reward as described in 3.4, and, since the observation is low dimensional, we do not use encoder ($\Phi(x) = x$). Technical details of the agents are listed in supplementary. Despite the simplicity of the environment, even a $4 \times 4$ labyrinth is challenging for the recurrent DQN agent. In this case, the random walk requires 518 steps on average to solve the task, giving rise to a long-term credit assignment problem for the recurrent DQN algorithm. As a result, the greedy policy shows worse performance with respect to a random walk. Our method addresses the problem. The error of the world model indicates rooms that have not been visited during the current episode, and stimulates the agent to learn to explore them. As a result, the agent inferred an exploration strategy, keeping track of visited rooms in the belief state.

## 4.2 Atari

We adopt the training and evaluation procedure of Kim et al. [19]. We train the LWM method on 6 hard exploration Atari [3] environments: *Freeway, Frostbite, Venture, Gravitar, Solaris* and *Montezuma's Revenge*. The training budget is 50M environment frames, the final scores averaged

Table 2: Final cumulative rewards for Atari. Listed results, except our LWM, are reported in [19].

|  | EMI [19] | EX2 [10] | ICM [25] | RND [6] | AE-SimHash [33] | LWM |
|------|----------|----------|----------|---------|-----------------|-----|
| Freeway | **33.8** | 27.1 | 33.6 | 33.3 | 33.5 | 30.8 |
| Frostbite | 7002 | 3387 | 4465 | 2227 | 5214 | **8409** |
| Venture | 646 | 589 | 418 | 707 | 445 | **998** |
| Gravitar | 558 | 550 | 424 | 546 | 482 | **1376** |
| Solaris | 2688 | 2276 | 2453 | 2051 | **4467** | 1268 |
| Montezuma | 387 | 0 | 161 | 377 | 75 | **2276** |

over 128 episodes of an $\epsilon$-greedy agent with $\epsilon = 0.001$, each experiment is performed with 5 different random seeds. One experiment requires 7.5h of a virtual machine with one Nvidia T4 GPU. Each environment has different magnitude of the reward values, we use intrinsic reward scaling $\beta = 0.01$ for *Freeway* and $\beta = 1$ for others. Results are listed in Table 2. Our method demonstrates an increase in performance with respect to prior work in 4 environments out of 6. It is noteworthy that the method has significantly improved the score of the notorious *Montezuma's Revenge*.

## 5    Conclusion

The simple tabular environment Partially Observable Labyrinth has demonstrated that the problem of partial observability with sparse rewards is challenging. We have employed a world model to indicate states required to explore addressing the problem. We have introduced the representation learning method for image-based observations, which arranges representations in the latent space according to the temporal distance of observations with W-MSE loss. We have proposed the LWM method, a predictive model of this latent space, to tackle the exploration problem for image-based environments and have demonstrated the performance on the Atari benchmark.

#### Acknowledgments

We are grateful to Aliaksandr Siarohin and Enver Sangineto for insightful discussions that helped writing this paper. We thank Google Cloud Platform (GCP) for providing computing support.

#### Broader Impact

The presented work is a research in the field of reinforcement learning, focusing on the problem of exploration in real-world conditions (image-based observations, partial observability). Such algorithms can help searching for new important information, for non-trivial solutions. These algorithms can be the crucial component for the development of autonomous intelligent systems for solving complex tasks. Such systems can be used in many different fields, having both strong positive and negative impacts on society, and should be treated with care.

#### Funding Disclosure

Funding in direct support of this work: Ph.D. scholarship by the University of Trento, Italy.

## Footnotes

[1]"Spherical distribution" denotes a distribution with a zero-mean and an identity-matrix covariance.

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
