[Supplementary Material]

# Supplementary Material

## Ablation Study

Table 1: Ablation Study. Bold marks results surpassing LWM from Tab. 2 of the manuscript. Experiments are performed with one seed.

|  | Feed-forward | Random Features |
|---|---|---|
| Freeway | **33.0** | 30.4 |
| Frostbite | 6353 | 5178 |
| Venture | **1206** | 810 |
| Gravitar | 1094 | 1116 |
| Solaris | 715 | 903 |
| Montezuma | 397 | 0 |

Tab. 1 presents ablation experiments. First, we have replaced RNN with a feed-forward network, keeping the W-MSE representation. Next, we have replaced the W-MSE representation with Random Features, keeping the RNN model. For *Montezuma's Revenge*, in both cases and the final score does not exceed 400. We noticed that both components of LWM work collaboratively: while the representation is optimised to preserve temporal consistency, the sequential prediction approximates well the trajectory of the agent.

For the Partially Observable Labyrinth, we experimented with the popular theoretically justified method MBIE-EB [Strehl and Littman, 2008]. However, this method does not assume partial observability and is not applicable to this environment: we observed a degradation in performance ($-55.7$ for size $3 \times 3$, $-1000$ for $4 \times 4$). Replacing the RNN world model with a feed-forward network produces an even more dramatic decline to $-969$ for size $3 \times 3$. In this case, the irrelevant intrinsic reward completely obscures the target goal.

## Noisy TV experiment

For an intuition about W-MSE representation and stochasticity, let's consider the noisy TV experiment: there is a TV in the environment, an agent can switch channels, but it always shows random images or noise. Observing it, most of the curiosity methods will produce an harmful high intrinsic reward, this effect being known as the "couch-potato" issue [Savinov et al., 2018]. In our case, the W-MSE loss pushes the representations of neighbour frames to be as similar as possible, thus the representations of random images of the TV will converge to the mean and will be easily predictable by the world model, avoiding the described issue.

## Algorithms

Alg. 1 represents high-level training scheme, Alg. 2 represents the intrinsic reward computation scheme.

## Hyperparameters

Tab. 2 represents the Convolutional Neural Network (CNN) architecture used for the encoder $\Phi$ and for DQN. Tab. 3 represents the details of $\Phi$ training. Tab. 4 lists pre-processing elements of Atari environments. Tab. 5 and Tab. 6 correspond to parameters of DQN and LWM. Recurrent Neural Networks (RNN) are GRU. FC denotes Fully Connected. Nonlinearities are ReLU. DQN target Q-network is updated every step with an exponentially moving average with a smoothing constant $\tau = 0.005$.

Parameters of Partially Observable Labyrinth experiments are presented in Tab. 7 and Tab. 8.

## Training dynamics

Fig. 1 shows training dynamics. For *Montezuma's Revenge*, the 2500 score was reached first time at 24.5M, 43.2M, 33.6M, 28.2M, 41M frame for each seed respectively.

**Algorithm 1:** Training cycle

**input** :buffer; encoder $\Phi$; network LWM; network DQN

$h \leftarrow 0$;                                                              /* DQN hidden state */
**while** *training* **do**
    Query two *recent* steps from buffer;
    Compute intrinsic reward $r_{recent}^{in}$ for *recent*;
    $action, h \leftarrow \text{DQN}(recent, r_{recent}^{in}, h)$;
    Step environment with $action$ and receive *output*;
    Append *output* to buffer;
    **if** *end of episode* **then**
        $h \leftarrow 0$;

    Sample *pairs* of observations from buffer;
    Update $\Phi$ with *pairs*;
    Sample *unrolls* from buffer;
    Compute intrinsic reward $r^{in}$ for *unrolls*;
    Update LWM with *unrolls*;
    Update DQN with *unrolls* and $r^{in}$;

---

**Algorithm 2:** Computation of the intrinsic reward for unroll

**input** :$unroll$; $dist_{mean}, dist_{std}$; $\beta$
**output** :rewards $r_i^{in}, i \in [1, N-1]$

$b \leftarrow 0$;                                                              /* LWM hidden state */
$o \leftarrow \text{Observations}(unroll)$;
$a \leftarrow \text{Actions}(unroll)$;
$N \leftarrow \text{Length}(unroll)$;
**for** $i \leftarrow 1$ **to** $N$ **do**
    $e_{i-1} \leftarrow \Phi(o_{i-1})$;
    $e_i \leftarrow \Phi(o_i)$;
    $pred, b \leftarrow \text{LWM}(e_{i-1}, a_{i-1}, b)$;
    $dist \leftarrow \text{MeanSquaredError}(e_i, pred)$;
    Update running average $dist_{mean}$ and $dist_{std}$ with $dist$;
    $r_i^{in} \leftarrow \frac{dist - dist_{mean}}{dist_{std}}$;
    $r_i^{in} \leftarrow min(max(r_i^{in}, -10), 10)$;
    $r_i^{in} \leftarrow \beta \times r_i^{in}$;

---

Table 2: CNN architecture

| | |
|---|---|
| Channels | 1, 32, 64, 64 |
| Kernels | 8, 4, 3 |
| Strides | 4, 2, 1 |

Table 3: $\Phi$ training with W-MSE loss

| | |
|---|---|
| Optimizer | Adam |
| Learning rate | $5 \cdot 10^{-4}$ |
| Pretrain iterations | 10000 |
| Batch size | 256 pairs |
| Input image size | $84 \times 84 \times 1$ |
| Output embedding size | 32 |
| Max spatial shift | 4 |
| Max temporal shift $L$ | 2 |

Table 4: Atari pre-processing

| | |
|---|---|
| Max episode length | 10000 steps |
| Action repeats | 4 |
| Frames stack | 1 |
| End episode on life loss | True |
| Reward clipping | False |
| Random noops range | 30 |
| Sticky actions | False |
| Frames max pooled | 3 and 4 |
| Grayscaled | True |
| Observation scaling | $84 \times 84$ |

Table 5: Atari DQN

| | |
|---|---|
| Optimizer | Adam |
| Learning rate | $10^{-4}$ |
| Adam epsilon | $10^{-3}$ |
| Clip gradient norm | 40 |
| Actors | 128 |
| Unroll | 80 steps |
| Burn-in | 40 steps |
| Batch size | 16 unrolls |
| N-step | 5 |
| Discount $\gamma$ | 0.99 |
| Target Q-network $\tau$ | 0.005 |
| Replay buffer size | $10^6$ |
| Replay warm-up | $4 \cdot 10^5$ |
| Replay priority exponent | 0.9 |
| Importance sampling exponent | 0.6 |
| Total environment frames | $5 \cdot 10^7$ |
| Training $\epsilon$ | $0.4^i, i \in [1, 8]$ |
| Evaluation $\epsilon$ | 0.001 |
| CNN output size | 512 |
| RNN input size | 512 + 1 + num. actions |
| RNN hidden size | 512 |
| Advantage FC layers | $512 \rightarrow 512, 512 \rightarrow$ num. actions |
| Value FC layers | $512 \rightarrow 512, 512 \rightarrow 1$ |

Table 6: Atari LWM

| | |
|---|---|
| Optimizer | Adam |
| Learning rate | $5 \cdot 10^{-4}$ |
| Pretrain iterations | 5000 |
| Mean and std momentum | 0.999 |
| FC layer before RNN | emb. size + num. actions $\rightarrow$ 128 |
| RNN input size | 128 |
| RNN hidden size | 256 |
| FC layers | $256 \rightarrow 256, 256 \rightarrow$ emb. size |

Table 7: POL DQN

| | |
|---|---|
| Optimizer | Adam |
| Learning rate | $5 \cdot 10^{-4}$ |
| Adam epsilon | $10^{-3}$ |
| Clip gradient norm | 40 |
| Actors | 8 |
| Unroll | 32 steps |
| Burn-in | 16 steps |
| Actors to learner iteration ratio | 4 |
| Batch size | 32 unrolls |
| N-step | 1 |
| Discount $\gamma$ | 0.99 |
| Target Q-network $\tau$ | 0.05 |
| Replay buffer size | $10^{5}$ |
| Replay warm-up | $10^{4}$ |
| Replay sampling | Uniform |
| Total environment frames | $10^{6}$ |
| Max episode length | 1000 steps |
| Training $\epsilon$ | 0.01 |
| Evaluation $\epsilon$ | 0.01 |
| FC layer before RNN | $4 + 4 + 1 \rightarrow 32$ |
| RNN input size | 32 |
| RNN hidden size | 128 |
| Advantage FC layers | $128 \rightarrow 128, 128 \rightarrow 4$ |
| Value FC layers | $128 \rightarrow 128, 128 \rightarrow 1$ |

Table 8: POL LWM

| | |
|---|---|
| Optimizer | Adam |
| Learning rate | $5 \cdot 10^{-4}$ |
| Pretrain iterations | 1000 |
| Mean and std momentum | 0.99 |
| FC layer before RNN | $4 + 4 \rightarrow 32$ |
| RNN input size | 32 |
| RNN hidden size | 128 |
| FC layers | $128 \rightarrow 128, 128 \rightarrow 4$ |
| Output nonlinearity | Sigmoid |
| Intrinsic reward scale $\beta$ | 1 |

Figure 1: Cumulative rewards of the last actor (with lowest $\epsilon$) during the training for Atari environments. The line corresponds to the average over 5 seeds, the light-blue area corresponds to the minimum and maximum.