[Reviews · NeurIPS 2020]

Review 1

Summary and Contributions: This paper proposes a method for exploration which learns an embedding and provides an intrinsic reward based on the prediction error of that embedding. The embedding itself is a newly proposed method that arranges embeddings based on their temporal distance. The method is empirically evaluated in a simpler labyrinth domain as well as some difficult Atari domains. The performance of the agent is shown to be consistent with, or better than recent state of the art exploration methods.

Strengths: The paper is both clearly articulated and has a clear story about how both the embedding and intrinsic reward works. It clearly explains the embedding algorithm works and of particular note is a clear Figure 2 that visually shows the process and result of the embedding. Additionally Figure 3 shows how the embedding works in one of the Atari games, Montezuma’s Revenge. This clearly shows how the algorithm works and its effect on one of the domains tested. Showing empirical results on the Labyrinth domain was very nice. The domain helped to empirically show that even a domain as simple as this can be difficult for DQN.

Weaknesses: Perhaps the biggest weakness of the paper is in how many moving parts and ideas there are in it. It is tough to tease apart where the results are coming from. What effect is whitening having on exploration vs. the intrinsic reward vs. the network setup. Each of these different parts contribute some aspect to the performance of the agent and it would be helpful to know what the effect of each individually is in relation to the performance of the agent with all the moving parts included. Following from the above - due to the amount of space needed to explain the above results there is very little discussion about the empirical results. It would have been nice to have a longer discussion about the results and to speak to some of the behavior of the agent based on them.

Correctness: The empirical results in the paper were clear - two things that could make that section better: Running the algorithms compared in the atari environment in the labyrinth. One advantage of a smaller domain is that it allows for quicker experimental results. It would have been nice to see how the other algorithms compare to this one in that simple domain. Are most of them able to perform well in that domain, but don’t perform as well as LWM when the domain is scaled up to something like Atari? The number of runs for the Atari results was 5. As shown in Deep Reinforcement Learning that Matters (Henderson, 2019), two different sets of 5 seeds can have significantly different results. Were you able to avoid the effects of seeds on the results?

Clarity: The paper was clearly written and did a good job of both explaining the methods used as well as the empirical results.

Relation to Prior Work: The paper does a good job of reviewing previous work and placing their method among them. They reviewed work such as POMDPs, exploration and intrinsic motivation, and self-supervised and representation learning.

Reproducibility: Yes

Additional Feedback: Overall the paper was nicely written and explained. Things that I think could help improve the paper: Results for the other algorithms tested in the labyrinth domain. Discussion about the areas that each part of the system may be contributing to the final results. While it’s likely too late for ablation studies, having a discussion about this would be helpful. More than 5 runs for the empirical results, or some explanation about how the effects of seeds were mitigated.


Review 2

Summary and Contributions: This paper uses a recently proposed self-supervised representation learning objective to encourage the formation of a latent representation of the world that has the property that successively encountered observations map to latent representations that have low L2 distance after a mini-batch whitening transformation. (The whitening transformation is one the elements that helps prevent the representation learning scheme from collapsing to a point.) The proposed approach then trains a world model on top of this self-supervised latent, learning to predict latent dynamics conditioned on actions. Prediction error from the latent world model is used to relabel trajectories from a replay buffer with an additional intrinsic reward, based on world-model prediction error, that encourages incentivises exploration in parts of the environment where the prediction error is high. The augmented trajectory fragments are used to train a standard DQN. Results are presented for toy-domain (a partially observed labyrinth), as well as for 6 Atari games.

Strengths: Intrinsic motivation, good exploration, and world models are all topics likely to interest the NeurIPS community that cares about RL. The method proposed in this work is simple, and seems to be effective. The two main elements leveraged are novel in combination, and are: (1) the particular self-supervised representation learning criterion (itself presented in a different paper currently still under review, but provided in the supplemental material); and (2) the use of a world-model based on this latent representation to update intrinsic rewards for trajectory fragments in the replay buffer (helping to make sure that the buffer is up to date, relative the the world-model’s recent perspective) The method appears to be helpful on both the toy labyrinth task (where no self-supervised representation is not used; just the retrospective re-labeling from the world mode), as well as on a collection of 6 Atari games. The authors have enabled others to easily inspect the details of their work, and to build upon it, by providing a dockerfile and pytorch source code to allow the experiments to be rerun.

Weaknesses: I would like to have seen a more thorough empirical investigation to understand the contributions of the different elements proposed, as well as other design choices. I believe the basic claims and methodology are correct.In its current form the paper presents several nice, simple ideas, but does an insufficient job at analysis and evaluating them in comparison with other methods.

Correctness: I believe the basic claims and methodology are correct.

Clarity: The paper is mostly clear, but the clarity could be improved in parts.

Relation to Prior Work: The paper does a reasonable job at situating itself in regard to other related work, however this could be significantly improved. In particular, there is a strong recency bias to the citations, and the opportunity to connect with older work from RL dealing with state prediction for representation learning is missed (e.g. PSR Singh et al 2004, Recurrent Predictive State Policy Networks Hefny et al 2018). Another connection that seems important to make regarding the proposed representation learning approach is Slow Feature Analysis (Wiskott, L. and Sejnowski, T.J., 2002. Slow feature analysis: Unsupervised learning of invariances. Neural computation, 14(4), pp.715-770.)

Reproducibility: Yes

Additional Feedback: This paper contains several nice ideas that seem to work well in combination. If this were a journal submission, I would be replying with an enthusiastic "revise and resubmit" recommendation. In particular, I would like to see a more extensive evaluation and comparison with other methods. Given the code available with the submission, I expect at least some of these would be easy for the authors to do. - It would be informative to see how the latent-world model performs on top of different feature learning approaches in a like-for-like setting in place of the current whitening scheme. (Since RF, IDF and CPC|Action are already available -- these might be quick tests to run.) - It would also help add weight to the paper if the authors were able to compare performance at a larger step-count. Though given the resources required, this may reasonable to leave to future work. I would also appreciate a more thorough treatment of related work.


Review 3

Summary and Contributions: EDIT: I really appreciated your rebuttal. It addressed a surprising number of my concerns in a very short space. I implore you to carry out the promised revisions in the final version. I have faith that you will do so, and as such I've raised my score from a 6 to a 7. The authors propose an intrinsic reward based on the error is a latent forward model. While this has been proposed before (cited in the paper), the novelty lies in the unique way the latent space is defined. The latent space is trained to push representations of temporally close states together, with batch whitening used to prevent degeneracy. The author's show that the error of the forward model yields a useful intrinsic motivation the empirically improves performance in hard exploration games.

Strengths: The authors do a great job thoroughly discussing related work and also spend a great deal of time illustrating their method. The T-SNE are the most convincing I've seen in some time.

Weaknesses: Two ablations would help clarify the contribution significantly. 1) Claims about episodic vs lifelong uncertainty are brought up and presumably the former is addressed by the LSTM in the LWM. But the utility of this is never tested -- a version of a feed-forward LWM would be appreciated. 2) As the representation learning objective is what sets this method apart from related work, an alternative objective in the same code base would strengthen the empirical results e.g. IDF or any of the other objectives mentioned in Figure 3. You do show ICM results, but they are from a different paper / codebase. As many implementation-specific assumptions often play a large role in performance, this would still be helpful. Some discussion of limitations is needed. Your results are impressive on the 50M regime, but others (e.g. NGU, RND) have shown much higher asymptotic performance. I don't expect you to run for billions of frames, but putting your results in this context would be appreciated. Given that DRL tends to be much more data efficient in the single actor setting, I'd even hypothesize that NGU might even be more performant in the 50M regime -- I'd either address that possibility with your own NGU experiments or acknowledge it as future work.

Correctness: The empirical methodology appears sound and in line with prior work. That said, Figure 1 in the Appendix seems likely to be incorrect. The min/max shading is likely incorrect, as otherwise the Montezuma's Revenge results imply that all 5 seeds got to their high score at exactly the same time, and generally have tightly coupled variance across seeds. The is likely just a plotting error -- I've encountered this before, but would be nice to correct it and confirm that it doesn't change any results.

Clarity: For the most part it is. However, aspects of the algorithms only become clear by referring to the appendix. And even then, it's still a bit unclear whether or not the embedding network is updated after the pretraining period, and whether this period is accounted for in the 50M observation budget. Also, the intuition for why the whitening objective deals with stochasticity is quite unclear compared to under work, like the IDF objective in ICM.

Relation to Prior Work: Yes, this aspect of the paper is quite strong.

Reproducibility: Yes

Additional Feedback: I think the representation learning objective should be mentioned more prominently. Right now the title could just as easily apply to several other papers that use forward models in a latent space for intrinsic rewards. Perhaps "Whitened Latent World Models..."? This would help in communicating this work to others and add some initial clarity.


Review 4

Summary and Contributions: The paper proposes a novel method to the address the problem of exploration in RL. It is know problem in RL that sparse rewards make random exploration _very_ inefficient. One approach for overcoming such limitations is using intrinsic motivation methods, building an auxiliary reward signal to encourage an agent to seek novel or rare states, for example proportional to inverse visit counts or, as proposed in this paper, some prediction error. Prediction error as a measure of novely can by heaviliy affected by three types of uncertainty by sources: 1. from novelty (epistemic) -- this is the signal we are typically after. 2. from partial observability --- to address this the agent must typically maintain a belief state for the environment. 3. from stochasticity (aleatoric) -- variations that irrelevant for exploration. This propose a belief state formulation that the authors claim is not too sensitivity to stochasticity and has the ability to extrapolate the state dynamics, such that the prediction error can be a genuine measurement for novelty. The paper builds on the following hypothesis: * temporal distance between observations is a good indication of how different (and possibly novel) they are * if we can successfully minimize distance between latent representations of nearby frames, the "stochastic elements will compensate each other, and at the same time, global trends will emerge." The paper proposes a self-supervised representation learning method based on the minimization of Euclidean distance between latent representations of temporally close observations, compound with a feature whitening transformation to numerically stabilize the learning process. The learnt latent representation is designed specifically for the exploration task, and the model is used directly to estimate the prediction error. Critically, the resulting prediction error signal is _not_ generated in pixel space, but instead calculated from the low-dimensional latent states, without relying on generative decoding of observation images. When summing the environment (extrinsic) reward with the intrinsic reward proportional to the prediction error of the world model, the resulting reward signal yields increased performance over baselines using prior exploration methods on several hard exploration Atari environments with sparse rewards.

Strengths: I would like to thank the authors of this submission - this is a paper that reads extremely well. The authors make an effort in submitting a polished paper, leading the reader very clearly from section to section. The authors state very clearly what the assumptions they are making, discussing them and explain how they will validate them, and what the contributions of the paper are. Nonetheless, I have some concerns, that I will list in the following section and that I hope the authors will be able to address in the rebuttal.

Weaknesses: * The authors in many places talk about capturing 'uncertainty', but only ever use MSE / prediction error as proxy, which in general is not a good measure of _uncertainty_. For example, how will the model capture distributions over states? The authors do not discuss (or test) their method in a stochastic environment, where performance will most likely be penalized (thi is on expected limitation, I guess). Given the authors specifically claim to propose a method to estimate uncertainty and novelty in the environment with the world model, I don't think the claim of capturing _uncertainty_ is sufficiently substantiated experimentally. * Why is temporal distance between observations a good indication of how different (and possibly novel) they are? The authors do not discuss when and how this assumption fails, and what the implications are. * The fisrt batch of experiments with the Labyrinth tasks is very interesting. In some sense I think it shows more how certain reward structures can be extremely challenging for credit assignment, and how injecting an intrinsic reward favoring exploration can troubleshoot learning, as opposed to showing anything particularly for the proposed method. How would other baselines behave in this setting? As a side note, in my experience R2D2 is typically very slow and tuned for achieving amazing performance over _very_ long training - I suspect this experiment would be more interesting with a faster agent, like a2c / IMPALA? * Whilst it is a bit unfortunate the the Atari experiments could run only to 50M frames (it would have been great to compare to the baselines in Badia et al), not everybody can run training for billions of frames. :( This will NOT be held against you.

Correctness: A few comments: * Line 159: "with a linear layer (i.e., the projection layer) to 32 dimensions, serving as an information bottleneck." Why _information_ bottleneck? * Line 156: "the stochastic elements will compensate each other" Why would they compensate each other? How do you test this? It is up to the author to show that this is the case. * Line 188: "400k observations of a random agent in Montezuma’s Revenge environment" in Figure 3 that is _clearly_ not a random agent. Can you please explain what we are seeing? * Generally speaking, if a T-SNE plot shows something interesting, it is an indication that 'something' is there, conversely, it T-SNE is not showing soemething that does not imply that the signal is not there. Therefore, I don't think we cannot really use T-SNE to solidly compare across methods. On the other hand, Figure 3 indeed indicates that W-MSE is clusering using temporal information - it's particularly cool to see how other properties are also preserved, for example 2 and 6 are close, and in both of them the character is in the middle of the frame.

Clarity: I think the paper reads very well and it clear. One comment: Line 115: 'it uses inverse dynamics features to estimate the similarity between observations' I would spend a few more word to clarify what actually is happening in more detail for folks not familiar with the work.

Relation to Prior Work: Some missing related work: Line 99: "The method was not demonstrated on image-based observations, as a suitable model of a complex environment is computationally infeasible." I can think of at least a couple of works that attempt just that: Hanfer et al, Dream to Control: Learning Behaviors by Latent Imagination Rezende et al, Causally Correct Partial Models for Reinforcement Learning

Reproducibility: Yes

Additional Feedback: Line 77: plese refrain from referring to agents as 'he'. --- AFTER REBUTTAL Thanks for the thorough rebuttal. Updating my score to 7.

[Author Response · NeurIPS 2020]

We thank the reviewers for thoroughly commenting on our article; their comments give us the opportunity to improve the manuscript and clarify all the unclear points. We have performed the additional requested experiments: 1) we have replaced the W-MSE representation with Random Features, keeping the RNN model; 2) we have replaced RNN with a feed-forward network, keeping the W-MSE representation. For Montezuma's Revenge, the average prediction error is significantly higher in both cases and the final score does not exceed 400, which matches the RND results from Table 2. We noticed that both components work collaboratively: while the representation is optimised to preserve temporal consistency, the sequential prediction approximates well the trajectory of the agent. These results will be included in the revised version.

We did not include the recent methods as baselines for the Partially Observable Labyrinth experiment because they were mostly designed for high-dimensional observations and do not provide benefits for this environment compared to the theoretically justified methods for the tabular setting. At the same time, to the best of our knowledge, popular classical methods for exploration do not assume partial observability and are not applicable to this environment. We experimented with the MBIE-EB method [30 in manuscript] and we observed a degradation in performance (-55.7 for size 3x3, -1000 for 4x4). Replacing the RNN world model with a feed-forward network produces an even more dramatic decline to -969 for size 3x3. In this case, the irrelevant intrinsic reward completely obscures the target goal.

The world model prediction error can be a proxy on how much information the belief state is missing about the predicted future step. The less information is available about this step, the more uncertain the model and the higher the error. During experiments with Partially Observable Labyrinth, we noticed that the world model tends to output mean values of the state when the observation cannot be predicted; this can be an indicator of uncertainty. However, as noted by R4, in general we cannot guarantee that the prediction error is a measure of uncertainty. More advanced methods, e.g. based on ensembles [1], can provide the estimation and can be a good extension of our method. To be precise, we will replace "uncertainty" with "missing information" in lines 37-40 and all other places.

For an intuition about W-MSE representation and stochasticity, let's consider the noisy TV experiment: there is a TV in the environment, an agent can switch channels, but it always shows random images or noise. Observing it, most of the curiosity methods will produce an harmful high intrinsic reward, this effect being known as the "couch-potato" issue [26 in manuscript]. In our case, the W-MSE loss pushes the representations of neighbour frames to be as similar as possible, thus the representations of random images of the TV will converge to the mean and will be easily predictable by the world model, avoiding the described issue.

As suggested, for future work we plan to check the scalability of the method and the asymptotic performance in Atari and compare it with the best-performing methods such as NGU. Furthermore, we plan to run experiments in 3D maze-like environments (e.g. as in [26 in manuscript]), which should be suitable for the method. Additionally, we plan to run the described noisy TV experiment. Predictive State Representations is an interesting, theoretically grounded alternative to the world models; we will include this missing citation. Our approach has conceptual similarities with Slow Feature Analysis; in fact, the recently presented gradient-based version of the method [2] can be a good alternative to the W-MSE loss; we will include this citation as well.

To show how the seed affects the performance we included Fig. 1 with training dynamics in the supplementary. However, as was accurately spotted out by R3, there is a mistake in the Montezuma's Revenge plot, i.e., it's a bug of the plotting script. We examined each run separately, the 2500 score was reached first time at 24.5M, 43.2M, 33.6M, 28.2M, 41M frame for each seed respectively; the plot will be updated in the final version. The encoder is updated during the whole training; empirically, the representation adapts to the changing distribution of observations. Pretraining steps are included in the training budget, these observations are used to train all components including DQN; we will add these details in the Method section. We will remove the "information bottleneck" phrase on line 159, as it can be misleading. Figure 3 shows the trajectory created from the user input ("emb_vis" script); it is used only for demonstration, while the representation is trained only on random transitions, it will be noted in the final version. We will elaborate on line 115. We will rephrase the line 99, as we refer to the specific model suitable for the VIME method [16 in manuscript]. We will refer to the agent as "it" on line 77 and other places.

Additionally, we would like to update the "Broader Impact" section in the revised version of the manuscript with: "The presented work is a research in the field of reinforcement learning, focusing on the problem of exploration in real-world conditions (image-based observations, partial observability). Such algorithms can help searching for new important information, for non-trivial solutions. These algorithms can be the crucial component for the development of autonomous intelligent systems for solving complex tasks. Such systems can be used in many different fields, having both strong positive and negative impacts on society, and should be treated with care."

[1] Balaji Lakshminarayanan, Alexander Pritzel, Charles Blundell. Simple and Scalable Predictive Uncertainty Estimation using Deep Ensembles, 2016. [2] Merlin Schüler, Hlynur Davíð Hlynsson, Laurenz Wiskott. Gradient-based Training of Slow Feature Analysis by Differentiable Approximate Whitening, 2018.


[Meta-Review · NeurIPS 2020]

All reviewers unanimously agree that this paper should be accepted to NeurIPS. The authors did a great job addressing almost all of the reviewer's concerns, leading to three reviewers increasing their score after the author response. Reviewers particularly praised the readability of the paper, the fact that the method is clearly defined, and that the authors did a good job of visually demonstrating how it works. However, the reviewers also agree that CPC|Action would be an important baseline to compare to, so I strongly encourage the authors to take the suggested improvements seriously and work towards an improved version of the paper. I am confident that the authors can make the requested changes and am recommending acceptance.